# Forecasting air pollution with deep learning with a focus on impact of urban traffic on PM10 and noise pollution

**Martin Kostadinov**[1]* , **Eftim Zdravevski**[1] , **Petre Lameski**[1] , **Paulo Jorge Coelho**[2,3] , **Biljana Stojkoska**[1] , **Michael A. Herzog**[4] , **Vladimir Trajkovik**[1]

**1** Faculty of Computer Science and Engineering, Ss. Cyril and Methodius University in Skopje, Skopje, N. Macedonia, **2** School of Technology and Management, Polytechnic of Leiria, Leiria, Portugal, **3** Institute for Systems Engineering and Computers at Coimbra (INESC Coimbra), DEEC, Coimbra, Portugal, **4** Magdeburg-Stendal University, Magdeburg, Germany

☯ These authors contributed equally to this work.
* martin.kostadinov@hotmail.com

**Data Availability Statement:** The data is available trough web sites of the government and meteorological stations data from Skopje and

## Abstract

Air pollution constitutes a significant worldwide environmental challenge, presenting threats to both our well-being and the purity of our food supply. This study suggests employing Recurrent Neural Network (RNN) models featuring Long Short-Term Memory (LSTM) units for forecasting PM10 particle levels in multiple locations in Skopje simultaneously over a time span of 1, 6, 12, and 24 hours. Historical air quality measurement data were gathered from various local sensors positioned at different sites in Skopje, along with data on meteorological conditions from publicly available APIs. Various implementations and hyperparameters of several deep learning models were compared. Additionally, an analysis was conducted to assess the influence of urban traffic on air and noise pollution, leveraging the COVID-19 lockdown periods when traffic was virtually non-existent. The outcomes suggest that the proposed models can effectively predict air pollution. From the urban traffic perspective, the findings indicate that car traffic is not the major contributing factor to air pollution.

## Introduction

Air pollution is the discharge of harmful pollutants into the atmosphere, impacting human health [1] and the global environment [2]. The World Health Organization (WHO) reports that every year, air pollution leads to nearly seven million fatalities worldwide. It is alarming that nine out of ten individuals breathe air surpassing WHO's recommended pollution levels, with the most severe consequences affecting residents of low- and middle-income nations. Predominantly, the primary source of air pollution stems from energy generation and consumption. The combustion of fossil fuels results in the release of gases and chemicals into the atmosphere. In a particularly detrimental cycle, air pollution not only contributes to climate change but is also exacerbated by it. Carbon dioxide and methane, as forms of air pollution, drive up global temperatures, intensifying another form of air pollution, namely smog. This

North Macedonia. We plan to publish the dataset in future publication after additional curation. The data used in the paper is available together with the code on the link: https://github.com/kmartin62/modeliranje-fuziranje/tree/master.

**Funding:** M.K., E.Z., P.L., B.S. and V.T. acknowledge that the work presented in this article was partially funded by the Ss. Cyril and Methodius University in Skopje, Faculty of Computer Science and Engineering. E.Z. acknowledges the support of NVIDIA through a donation of a Titan V GPU. M.K., E.Z., P.L., B.S., V.T. and M.A.H. acknowledge the support of the CleanBreathe project funded by Bundesministerium für Bildung & Forschung (01DS21018). P.J.C. acknowledges the funding by FCT/MEC through national funds and, when applicable, co-funded by the FEDER-PT2020 partnership agreement under the project (UIDB/00308/2020).

**Competing interests:** The authors have declared that no competing interests exist.

smog is exacerbated when weather conditions are warmer and ultraviolet radiation is more prevalent. Furthermore, climate change amplifies the production of allergenic air pollutants, such as mold, due to increased humidity caused by extreme weather events, increased flooding, and an extended pollen season due to changing climate patterns.

Some studies show that in addition to the previously mentioned particles, PM10 and PM2.5 [3] play a significant role in increasing the incidence of respiratory diseases. The situation in Skopje, the capital of Macedonia, is quite severe, particularly during the winter, and often ranked among the world's most heavily polluted cities [4]. In [4], authors performed extensive experiments related to the chemical initialization of forecasting models to forecast the weather and air quality over the country simultaneously. There have been other approaches for tackling this problem in Skopje, namely [5–9] that have investigated different deep learning methods and combining weather data and other multi-modal data. However, this study is the first approach that attempts to analyze the impact of traffic on air and noise pollution.

In recent years, we have witnessed an enormous number of innovations and developments in the field of information technology (IT). Among these technologies is the Internet of Things (IoT) paradigm, which has gained widespread popularity due to its capacity to deploy additional sensors in vital locations and gather extensive data. Moreover, cloud computing technologies are employed for the analysis and identification of patterns, as mentioned in a previous study [10], indicating potential pollution events, thereby enabling almost real-time monitoring and visualization. Support vector machine (SVM) models and artificial neural networks (ANNs) have found application in diverse prediction tasks, spanning business and financial domains [11] to a variety of environmental problems [12]. Predictive air pollution models are particularly useful because they can be used by governments and play a significant role in finding smarter strategies and preemptive actions to tackle this problem. This approach can help states reduce costs, increase accuracy, and detect future possibilities through probability. For this purpose, monitoring and forecasting air pollution has evolved into a valuable strategy in contemporary urban environments. This capability empowers governing bodies to observe pollution levels and determine when to implement preventive measures, including policies to reduce traffic, regulations for industrial facilities, closures of public spaces, and advisories aimed at minimizing exposure for vulnerable individuals, as noted in a prior study [13].

Fronza et al. [14] conducted research and demonstrated that air quality data can be characterized as stochastic time series, implying the feasibility of developing models for predicting future air pollutant levels. Various techniques, including models within the realm of statistical learning, machine learning, and deep learning, have proven successful in extracting patterns directly from input data and learning from the data's distribution. Additionally, it is noteworthy that alternative methods have effectively forecasted extensive multi-sensor datasets by employing sliding window feature extraction and ensemble selection. These methodologies can find application in areas with pollution concerns and beyond. Such advancements are of great significance, as taking preemptive actions can proactively reduce the influence on air quality to prevent or, at the very least, reduce citizens' exposure to detrimental air pollution.

This paper utilizes air quality measurements along with meteorological data to predict levels of air pollution in the urban region of Skopje for time intervals of 1, 6, 12, and 24 hours. The main motivation for this work is to confirm the possibility of using a multivariate dataset that contains both pollution and meteorological data for short-term PM10 pollution prediction. The primary innovation of this study lies in the utilization of diverse datasets obtained from local sensors to enhance prediction accuracy. Furthermore, historical data was compiled from meteorological stations in the vicinity of Skopje. The performance of various models employing LSTM networks and Convolutional Neural Networks (CNNs) was assessed and a comparative analysis of their effectiveness was conducted. Additional contribution is the newly

compiled and preprocessed dataset made publicly available with this study and the applied methods implementation and the presented results and the identification of the influence of traffic to the PM10 pollution in the area of Skopje.

The structure of this paper is outlined as follows: In Section Related work, the paper provides an overview of recent studies relevant to air pollution prediction. Section Materials and Methods outlines the materials and methods used in this study. The findings of the experiments are presented in section Results. Subsequently, section Discussion offers an in-depth analysis and discussion of the results. Finally, conclusions are presented in the Conclusion section.

## Related work

In recent times, the challenge of air pollution prediction has been addressed through diverse models, encompassing both conventional and deep learning methodologies. These models predominantly rely on Collecting measurements of various pollutants, including particulate matter (PM) such as PM2.5 and PM10, and gaseous components (e.g., $NO_2$, CO, $O_3$, and $SO_2$), obtained from sensors at specific time points and locations. Furthermore using meteorological data, these models have been enhanced by incorporating both present and predicted factors such as humidity, temperature, wind speed, and precipitation. [15, 16].

The study proposed using three distinct Multiple Linear Regression (MLR) models for forecasting PM10 concentrations for various time intervals [17]. Additionally, authors in [18], focused on forecasted the concentration of a heavily polluted area using Artificial Neural Networks (ANNs) with Real-Time Correction (RTC) were utilized to predict various air pollutants for both the present day and the following four days.

Another category of methods includes those that employ LSTM neural networks [19]. LSTM represents an artificial RNN architecture employed in the field of deep learning. [20]. LSTMs are acknowledged as a prevalent neural network architecture adept at managing sequences and lists, owing to their chain-like structure. These models have proven successful in forecasting tasks across diverse domains, including the prediction of financial time series [21], sensory data [22] and even cyber attacks [23]. VARIMA models [24] offer an alternative to auto-regressive techniques when incorporating multiple time series into the modeling task. However, unlike auto-regressive models that rely solely on linear relationships among features, LSTM provides an advantage by capturing non-linear interactions during the modeling process.

A comparable method was outlined in [6], where convolutional neural networks were integrated with LSTMs to categorize PM10 concentrations. In [25], a methodology for predicting air pollution utilizing an RNN with LSTM was introduced. Various models based on deep neural networks have since surfaced [25]. In a similar vein, techniques utilizing fully connected neural networks, including RNNs [26] and LSTM networks, have become prominent. Certain approaches also integrate CNNs to enhance overall performance of RNN-based air pollution prediction [27–29]. In addition, certain methodologies leverage autoencoder models [30], sequence-to-sequence models [31], neural networks that integrate linear predictors into ensembles, Bayesian networks, and multi-label classifiers [32]. Another intriguing approach was detailed in [33], where an attention-based model was adopted. In this approach, the attention mechanism was exclusively employed on the wind measurements to generate an encoded value, which was utilized as a data augmentation technique within the primary model.

## Materials and methods

The approach and method suggested for predicting air pollution, described in this article, utilize the architecture shown in Fig 1.

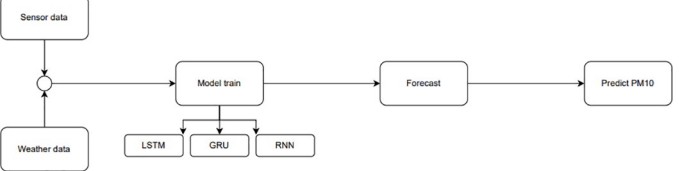

**Fig 1. Data flow of the proposed method for air pollution forecasting.**

## Data set and preprocessing

The dataset employed in this research comprises measurements from air quality sensors situated at various points throughout Skopje. Historical values of PM10 and PM2.5 particles were obtained using a publicly available API at fifteen-minute intervals. The dataset is enriched with meteorological parameters such as atmospheric temperature, humidity, and pressure, which were measured at each location where the pollution sensors were placed. The location is determined by the longitude and latitude properties obtained from the sensor information. The research assessed various sliding window durations, which included 1, 6, 12, and 24 successive measurements gathered from the air pollution sensors and meteorological stations.

The data for both air pollution and meteorological stations' measurements were collected every fifteen minutes. However, an issue with the inconsistency was encountered in the timing of the pollution data and meteorological data. While the API for meteorological data had the option to provide information precisely every fifteen minutes, the API for the pollution data gave information within a time span of a plus-minus five to ten minutes from the expected time. For instance, the meteorological station was able to provide data for 01.01.2020 at 00:15:00h, whereas the pollution API provided data for 01.01.2020 at 00:21:00h. As a solution, the pollution data was interpolated and synchronized with the meteorological data by time.

There are several types of interpolation, but for this problem, it was decided to use linear interpolation. Linear interpolation serves as a valuable technique when seeking a value between provided data points. It can be likened to 'filling in the gaps' in a data table. The concept behind linear interpolation involves drawing a straight line to connect the known data points that flank the unknown point.

An earlier investigation employing a comparable approach was detailed in [5], in which an attempt was made to forecast PM10 particle values 3 hours into the future. In [6], in a somewhat analogous approach, a prior study was outlined, wherein a portion of the data was utilized to forecast forthcoming values by combining LSTM and CNN. In contrast to those methodologies, this research additionally considers PM2.5 values and the concentrations of additional meteorological variables at the measurement stations in Skopje, which have not been examined previously. The locations in Skopje where the data was collected data are listed in Table 1.

The measurements gathered in each location are: Temperature, Noise, PM10, Absolute Humidity, PM2.5, Relative humidity, Wind speed, Station Pressure, Sea level pressure (station pressure corrected for sea level), Solar elevation angle, Solar radiation, Pressure, Snow (accumulated snowfall in mm), UV, Wind direction, Visibility (measured in kilometers), and Clouds (average total cloud coverage in percents).

To address disparities in sampling frequencies and time windows with missing values within the sensor and meteorological data, certain preprocessing steps were necessary. These preprocessing procedures were conducted on both the training and validation sets and included the following steps:

**Table 1. Municipalities and representative areas in Skopje, where the sensor data was collected.**

| Municipality | Area |
| --- | --- |
| Kisela Voda | 11 Oktomvri, Crniche, Shampionche |
| Gazi Baba | Avtokomanda, Zhelezara |
| Karposh | Bardovci, Nerezi, Zoo |
| Butel | Butel 1, Butel 2 |
| Drachevo | Drachevo Kuzman |
| Gjorche Petrov | Gjorche Stanica |
| Ilinden | Ilinden |
| Centar | Madjir Maalo |
| Aerodrom | MZT, Novo Lisiche, TC Skopjanka |
| Sopishte | Sopishte |

- Missing data interpolation;

- Cleaning outliers using the three-sigma rules;

- Min-Max normalization;

- Samples for data window preparation.

The dataset was partitioned into separate training, validation, and test sets. The training data encompassed the period from September 11, 2017, to February 3, 2022, amounting to a total of 151,199 samples. The validation samples were dynamically selected, constituting 20 percent of the training data points amounting to a total of 30,239 samples. Hyperparameter tuning was done manually on the data for November 11, 2019, following the approaches and methods in [7]. To expedite the hyperparameter tuning process, KerasTuner [34], a scalable hyperparameter optimization library was adopted. This facilitated the exploration of a vast hyperparameter space and the identification of the optimal configuration for the used models.

## The architecture of deep learning models

This article compares several diverse model architectures and their performance is analyzed. It was decided to use an RNN, which includes LSTM and SimpleRNN layers, and CNN layers to build the models, adding dropout layers in some of the architectures to address transient sensor failures. Additionally, the performance of gated recurrent units (GRU) was evaluated, as another representative of recurrent neural networks [35].

The reason for choosing these types of deep learning models is that the dataset contains time series data. RNNs have proven to be good at learning time series data characteristics and patterns, while CNNs are able to detect local features and patterns in the data even for time series. Both approaches are able to generate representative features from time series data and have proven to build models which are good at forecasting values.

Recurrent neural networks primarily excel at processing sequential data, such as text, speech, and frequently time series data. These data types exhibit a sequential relationship, where each data point is linked to its preceding counterpart. An illustrative example is meteorological data, where a day's temperature relates to the prior day's temperature or even the preceding hour. Consequently, numerous sequences can be derived from continuous data, allowing for the observation and analysis of correlations between these sequences. All models in this study were trained using a merged dataset that incorporates data from various PM10 air quality measurement stations and meteorological stations. Ultimately, the models'

effectiveness in making short-term predictions for time horizons of +1, +6, +12, and +24 hours into the future for all locations in the dataset was assessed.

Neural networks are used to reconstruct a function, as depicted in Eq 1, where $x_i$ is the input vector, $w_i$ represents the weights' matrix, and $g$ is the activation function.

$$y = g(\sum x_i * w_i) \tag{1}$$

Choosing an optimal input vector with the correct causality and correlation to the target feature poses a non-trivial challenge. Diverse combinations of features and various input vector sizes were investigated to determine which input features yielded the most favorable results. In [26], experiments were conducted using input vectors that varied from a single PM10 sensor value to a six-feature vector. Through the utilization of models incorporating LSTM and RNN layers, it was determined that the most effective approach for short-term air pollution prediction involved amalgamating PM10 values from all sensors with meteorological parameters. This vector encompassed a range of PM10 levels, temperature, and additional meteorological data. These input features served as the foundation, and potential enhancements were explored.

The additional aim was to diminish the mean squared error by broadening the dataset. To achieve this, PM2.5 level values were incorporated into the training dataset, collected by the same sensor used for PM10 forecasting. Notably, a substantial correlation between PM10 concentration and PM2.5 levels was observed. The extended dataset implemented in this experiment yielded improved results. However, introducing a second set of PM2.5 measurements from a different location did not yield further enhancements; in fact, it led to a decline in the LSTM model's performance. Throughout most of the experiments, the Rectified Linear Unit (ReLU) function was utilized as the activation function, shown in Eq 2:

$$ReLU(x) = \max(0, x) \tag{2}$$

and in this context, "x" represents the input signal to a neuron. Additionally, the results of different approaches were explored by executing experiments with different activation functions, such as tanh, the sigmoid function, and the Scaled Exponential Linear Unit (SELU), depicted in Eqs 3, 4 and 5:

$$f(x) = tanh(x) \tag{3}$$

$$sigmoid(x) = \frac{e^x}{e^x + 1} \tag{4}$$

SELU, short for Scaled Exponential Linear Unit, is a variant of the ReLU function, and it is defined as follows::

$$SELU(x) = \begin{cases} \lambda x & \text{if } x > 0 \\ \lambda(\alpha e^x - \alpha) & \text{if } x \leq 0 \end{cases} \tag{5}$$

In this context, $\lambda$ and $\alpha$ represent predetermined constants. The SELU addresses the vanishing gradient problem and was first introduced in [36].

For these experiments, the mean squared error loss function was employed, and for model optimization, the Adam optimizer was applied [37]. The experiments were implemented with TensorFlow, specifically using Keras [38].

## Parameter tuning

Additional effort was made to determine how some pre-tuned hyperparameters would perform and what kind of results would be obtained. For this reason, the hyperparameters from [7] were considered since these aimed to solve a similar problem to the one at hand, but with less meteorological data and fewer locations. Initially, these parameters were used on the dataset for 11 Oktomvri to determine which model presents the best results. This location was chosen because it is geographically located in the middle of all the other locations used in this paper.

The following parameter ranges were assesed to determine the most suitable architecture for this task:

- **Dropout**: Deep neural networks with a substantial number of parameters can offer significant capabilities, yet overfitting can pose a challenge, especially when working with relatively small datasets. In such cases, neural networks might struggle to generalize and make predictions for new data, given the limited control over the learning process. To combat this issue, dropout is employed as a technique. The concept behind dropout is to selectively deactivate units within the neural network during the training phase, thereby preventing excessive co-adaptation among units.

- **Learning rate**: The learning rate is a hyperparameter responsible for regulating the magnitude of adjustments made to the model with each weight update, guided by the estimated error. Selecting the appropriate learning rate is a challenging task, as a value that is too small can lead to protracted training, potentially getting the model stuck. Conversely, a learning rate that is too large can prompt swift but suboptimal weight updates, potentially destabilizing the training process. Essentially, the learning rate determines the pace at which the model adapts to the problem at hand.

- **LSTM layer units**: The number of LSTM cells in the layer acts as a parameter in the model optimization, and the dimensionality of the output space is determined by the number of units.

- **RNN units**: This pertains to the number of RNN cells within the layer. By default, an RNN layer generates a single vector per sample as its output. This vector encapsulates the output of the RNN cell at the final time step, containing information about the entire input sequence. The 'units' parameter controls the dimensions of this output. Alternatively, an RNN layer can also provide the entire output sequence for each sample, producing one vector for each time step in the sequence for every sample.

- **Convolutional kernel size**: The dimensions of the convolution window are determined by the size of the convolutional kernel. It functions as a filter mask during feature extraction and specifies both the height and width of the window.

- **Number of filters in the Conv1D layer**: This parameter defines the quantity of output filters in the convolution.

Table 2 indicates the parameters that were optimized for each of the proposed architectures.

## Feature analysis and selection

Feature selection is a crucial step in building effective and efficient machine learning models [39] when useing classical machine learning methods. The process entails identifying and preserving the most pertinent features from the dataset to enhance model performance while also

**Table 2. Overview of the assessed methods utilizing varied setups with regard to LSTM units, GRU units, dropout rate, and CNN kernel size.**

| # | Architecture | Parameters |
|---|---|---|
| 1 | LSTM + Dense | LSTM Units (64) + LR (0.0001) |
| 2 | LSTM + LSTM + Dense | LSTM Units (32, 64) + LR (0.0001) |
| 3 | GRU + Dense | GRU Units (64) |
| 4 | Conv1D + Flatten + Dense | Conv1D kernel size (64) |
| 5 | LSTM + Dropout + LSTM + Dense | LSTM Units (64, 32), Dropout (0.2) |
| 6 | SimpleRNN + LSTM + Dropout + LSTM + Dense | LSTM Units (64, 32), SimpleRNN (128), Dropout (0.2) |
| 7 | Conv1D + MaxPooling1D + Flatten + Dense | Conv1D kernel size (64), MaxPooling1D (2) |

reducing computational complexity. In this paper, three popular feature selection methods are explored: Pearson's correlation coefficient [40], Chi-squared test [41], and K best features [42] from Linear Regression, RandomForest Regression, and LightGBM.

The feature analysis in this case is performed to identify the importance of the features and gain additional insights in the data. The models used for the prediction are end-to-end models that do not necessarily benefit from feature selection, and as such, the feature analysis is not included in this process in the modeling.

Pearson's correlation coefficient is a statistical metric used to evaluate the linear association between two continuous variables. Its value falls within the range of -1 to +1, where -1 denotes a flawless negative linear correlation, +1 signifies a flawless positive linear correlation, and 0 indicates the absence of a linear correlation. This method is particularly useful for identifying feature pairs that exhibit strong linear relationships, which can help eliminate redundant or highly correlated features.

The Chi-squared test measures the independence between two variables in a contingency table. By comparing observed and expected frequencies, this method identifies features that have significant associations with the target variable. It is particularly useful in feature selection.

Additionally, the K best features approach from three popular regression algorithms are explored: Linear Regression, RandomForest Regression, and LightGBM. Linear Regression is a straightforward and easily interpretable regression model that aims to establish a linear connection between the input features and the target variable. By selecting the K best features based on their coefficients or p-values, the most influential predictors for the target variable can be extracted.

RandomForest Regression is an ensemble learning technique that builds numerous decision trees and combines their predictions to achieve more precise and resilient outcomes. The feature selection process in RandomForest Regression involves evaluating the importance of each feature based on how much they contribute to the reduction of impurity in the trees. This method helps in identifying the most relevant features that lead to better predictive performance.

LightGBM is another powerful ensemble learning technique based on gradient boosting. It differs from traditional gradient boosting algorithms by using a novel decision tree construction method and handling data in a more efficient way. LightGBM selects important features by calculating the split gain or feature importance during the tree-building process, allowing us to retain only the most impactful features for the model's predictive power.

By exploring and applying these feature selection methods from different angles, the goal is to identify the most relevant and informative features for the machine learning model, ultimately enhancing its performance and interpretability.

## Impact of urban traffic on air and noise pollution

To investigate the impact of urban traffic on air and noise pollution, the lockdown periods during the COVID-19 pandemic, when there was practically no urban traffic, were utilized.

During the analysis, a portion of the dataset ranging from March 15 to May 20, 2020 was selected and labeled as the "COVID period". The three months prior, from January 1 to March 1, were designated as the "Pre-COVID period", and the three months following were labeled as the "Post-COVID period" for the year 2020. In addition, another segment was designated as the "Last year quarantine period", which included the period from March 15 to May 20, for the year 2019. This research included five locations: 11 Oktomvri, Bardovci, Crniche, Gjorche Stanica, and Zhelezara, which cover a significant portion of the city of Skopje.

The described periods were selected, and a comparison of the values of PM-10 particles and noise levels was conducted. Pearson's correlation coefficient, the most common method for quantifying a linear correlation, was employed to assess the influence of noise on PM10 particles. This coefficient, ranging from -1 to 1, gauges the magnitude and direction of the association between two variables. When one variable undergoes a change, the other variable experiences a concurrent change in the same direction. The conclusion from this method will tell us whether there is an impact on the pm10 particles from the noise i.e. whether there is a linear dependence between these two. A linear dependence would mean that increasing the noise should increase the pm10 particles which would imply that the noise is a factor for increasing the pm10 and therefore the air pollution.

## Results

### Methods evaluation

The impact of the activation function on the model's performance was initially examined. The tables in this section display distinct Mean Squared Error (MSE) and Root Mean Squared Error (RMSE) results obtained with different architectures and activation functions on the validation set for time windows of +1, +6, +12, and +24 hours, utilizing the '11 Oktomvri' dataset. MSE and RMSE are among the most commonly used evaluation metrics for regression models, as noted in recent systematic literature reviews [43]. Their sensitivity to large errors, mathematical properties, interpretability, and ability to capture overall model performance make them excellent choice for assessing the accuracy and reliability of regression models. By using these metrics, the models not only fit the data well but also generalize effectively to new, unseen data.

$$\text{MSE}(y, \hat{y}) = \frac{\sum_{i=0}^{N-1} (y_i - \hat{y}_i)^2}{N} \tag{6}$$

$$\text{RMSE}(y, \hat{y}) = \sqrt{\frac{\sum_{i=0}^{N-1} (y_i - \hat{y}_i)^2}{N}} \tag{7}$$

The optimal activation function differed depending on the model's architecture. Therefore, the most appropriate activation function for the respective architectures was used in subsequent experiments, as outlined in the tables below.

From the Table 3 analysis, the model with the best performance and lowest MSE was obtained using GRU and a Dense layer using the tanh activation function. The performance of this model for different time horizons can be observed in Figs 2–5.

**Table 3. Summary of the explored methodologies alongside their average forecasting performance, evaluated in MSE and RMSE for the 1-hour time horizon.** The superior performance is highlighted in bold.

| Architecture | Score | ReLU | tanh | sigmoid | SELU |
|---|---|---|---|---|---|
| LSTM + Dense | MSE | 0.0682 | **0.0681** | **0.0681** | 0.0683 |
| | RMSE | 0.2611 | 0.2609 | 0.2610 | 0.2614 |
| LSTM + LSTM + Dense | MSE | **0.0683** | 0.0686 | 0.0686 | 0.0684 |
| | RMSE | 0.2614 | 0.2620 | 0.2619 | 0.2615 |
| GRU + Dense | MSE | 0.0682 | **0.0679** | 0.0680 | 0.0680 |
| | RMSE | 0.2611 | 0.2606 | 0.2607 | 0.2608 |
| Conv1D + Flatten + Dense | MSE | 0.0769 | 0.0789 | **0.0754** | 0.0838 |
| | RMSE | 0.2773 | 0.2809 | 0.2746 | 0.2895 |
| LSTM + Dropout + LSTM + Dense | MSE | **0.0700** | 0.0702 | **0.0700** | 0.0704 |
| | RMSE | 0.2646 | 0.2650 | 0.2645 | 0.2654 |
| SimpleRNN + LSTM + Dropout + LSTM + Dense | MSE | **0.0718** | 0.0720 | 0.0719 | 0.0720 |
| | RMSE | 0.2680 | 0.2684 | 0.2682 | 0.2682 |
| Conv1D + MaxPooling1D + Flatten + Dense | MSE | 0.1080 | **0.1026** | 0.1079 | **0.1026** |
| | RMSE | 0.3286 | 0.3203 | 0.3285 | 0.3203 |

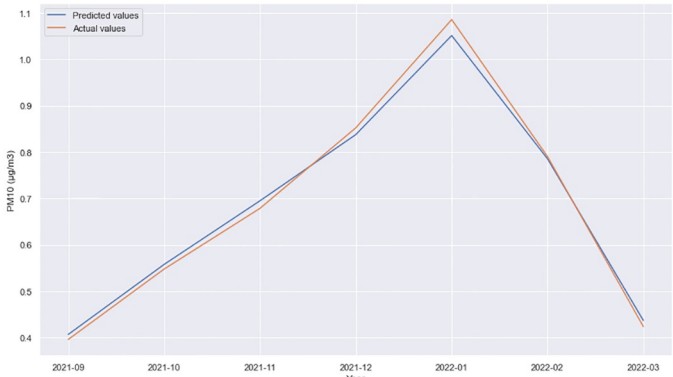

**Fig 2. GRU + Dense layer, forecast comparison for 11 Oktomvri validation data set with the 1h time horizon.**

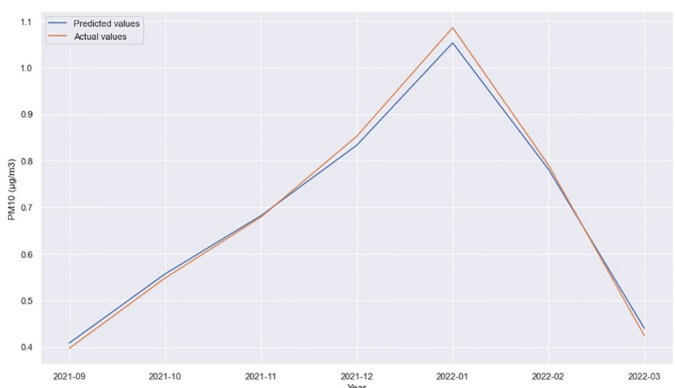

**Fig 3. GRU + Dense layer, forecast comparison for 11 Oktomvri validation data set with the 6h time horizon.**

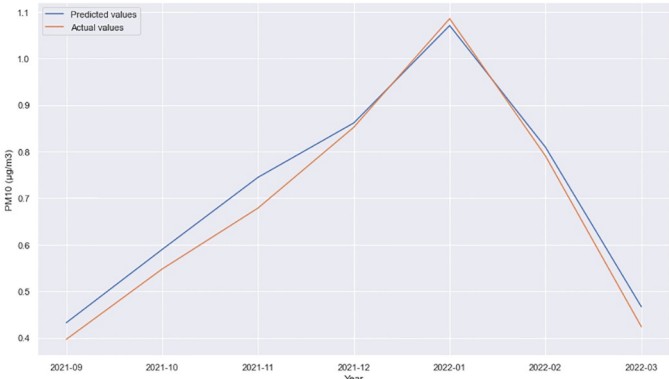

**Fig 4. GRU + Dense layer, forecast comparison for 11 Oktomvri validation data set with the 12h time horizon.**

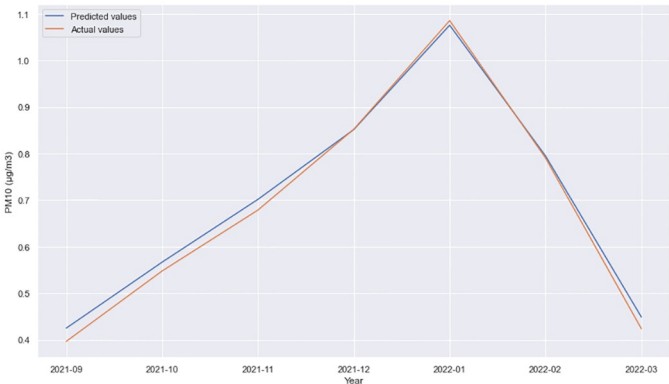

**Fig 5. GRU + Dense layer, forecast comparison for 11 Oktomvri validation data set with the 24h time horizon.**

For the 6h horizon, Table 4 depicts the performances of the algorithms. The best performance and lowest MSE were obtained by combining GRU with a Dense layer using tanh as an activation function.

The best performance and lowest MSE are obtained by GRU combined with a Dense layer using ReLU as an activation function, for the 12h time horizon, as depicted in Table 5.

Again, the best performance and lowest MSE are obtained by GRU in combination with a Dense layer using sigmoid as an activation function, as presented in Table 6.

As presented from the previous results, GRU in combination with a Dense layer provided the best results. Therefore, the same model was used for multivariate regression to forecast PM10 levels for all locations.

## Multivariate results

This subsection discusses the multivariate regression in which we forecasted PM10 for all locations, as presented in the experiments depicted in Figs 6–9. The experiment was conducted in order to see how is one model with multiple outputs going to behave and whether technology costs for production can be reduced, for example, costs for model deployment, model validation etc. Also, the maintenance of one single model is handled better than maintaining six different models for each location.

**Table 4. Summary of the evaluated methodologies along with their respective average forecasting performance, assessed in terms of MSE and RMSE for the 6-hour time horizon.** The superior performance is highlighted in bold.

| Architecture | Score | ReLU | tanh | sigmoid | SELU |
|---|---|---|---|---|---|
| LSTM + Dense | MSE | 0.0675 | **0.0677** | 0.0679 | 0.0678 |
| | RMSE | 0.2599 | 0.2601 | 0.2606 | 0.2604 |
| LSTM + LSTM + Dense | MSE | **0.0675** | 0.0677 | 0.0671 | 0.0680 |
| | RMSE | 0.2597 | 0.2602 | 0.2590 | 0.2607 |
| GRU + Dense | MSE | 0.0671 | **0.0669** | 0.0677 | 0.0672 |
| | RMSE | 0.2590 | 0.2602 | 0.2587 | 0.2592 |
| Conv1D + Flatten + Dense | MSE | **0.0722** | 0.1610 | 0.1125 | 0.1154 |
| | RMSE | 0.2687 | 0.4012 | 0.3354 | 0.3397 |
| LSTM + Dropout + LSTM + Dense | MSE | **0.0689** | 0.0698 | 0.0691 | 0.0693 |
| | RMSE | 0.2625 | 0.2641 | 0.2629 | 0.2633 |
| SimpleRNN + LSTM + Dropout + LSTM + Dense | MSE | **0.0709** | 0.0712 | 0.0710 | 0.0760 |
| | RMSE | 0.2662 | 0.2668 | 0.2665 | 0.2757 |
| Conv1D + MaxPooling1D + Flatten + Dense | MSE | 0.1080 | **0.1026** | 0.1079 | **0.1026** |
| | RMSE | 0.3286 | 0.3203 | 0.3285 | 0.3203 |

**Table 5. Summary of the evaluated methodologies along with their respective average forecasting performance, assessed in terms of MSE and RMSE for the 12-hour time horizon.** The superior performance is highlighted in bold.

| Architecture | Score | ReLU | tanh | sigmoid | SELU |
|---|---|---|---|---|---|
| LSTM + Dense | MSE | 0.0673 | 0.0675 | 0.0676 | **0.0671** |
| | RMSE | 0.2594 | 0.2599 | 0.2600 | 0.2590 |
| LSTM + LSTM + Dense | MSE | 0.0673 | 0.0673 | **0.0668** | 0.0672 |
| | RMSE | 0.2593 | 0.2595 | 0.2584 | 0.2592 |
| GRU + Dense | MSE | **0.0666** | 0.0669 | 0.0677 | 0.0675 |
| | RMSE | 0.2581 | 0.2586 | 0.2601 | 0.2597 |
| Conv1D + Flatten + Dense | MSE | 0.4999 | 0.1704 | 0.1013 | **0.0730** |
| | RMSE | 0.7070 | 0.4128 | 0.3183 | 0.2702 |
| LSTM + Dropout + LSTM + Dense | MSE | **0.0686** | 0.0690 | 0.0690 | 0.0691 |
| | RMSE | 0.2619 | 0.2628 | 0.2626 | 0.2629 |
| SimpleRNN + LSTM + Dropout + LSTM + Dense | MSE | 0.0706 | 0.0707 | **0.0700** | 0.0710 |
| | RMSE | 0.2657 | 0.2660 | 0.2645 | 0.2664 |
| Conv1D + MaxPooling1D + Flatten + Dense | MSE | **0.1001** | 0.1821 | 0.1341 | 0.1041 |
| | RMSE | 0.3163 | 0.4267 | 0.3661 | 0.3227 |

Table 7 summarizes the evaluated approaches and their performance.

## Features selection

Multiple algorithms were used to obtain more insights about the importance of each of the features used, and feature selection was performed to achieve better results. All of the features used in the dataset are numerical. In addition to Pearson's correlation coefficient, the chi-squared test was used, a method known as recursive feature extraction. Furthermore, the K best features were selected from LinearRegression, RandomForestRegressor, and LightGBM, which is a gradient boost model. The results showed that PM2.5 has the greatest impact on

**Table 6. Summary of the evaluated methodologies along with their respective average forecasting performance, assessed in terms of MSE and RMSE for the 24-hour time horizon.** The superior performance is highlighted in bold.

| Architecture | Score | ReLU | tanh | sigmoid | SELU |
|---|---|---|---|---|---|
| LSTM + Dense | MSE | 0.0673 | 0.0673 | **0.0668** | 0.0672 |
| | RMSE | 0.2593 | 0.2595 | 0.2584 | 0.2592 |
| LSTM + LSTM + Dense | MSE | 0.0668 | 0.0669 | **0.0666** | 0.0669 |
| | RMSE | 0.2585 | 0.2586 | 0.2581 | 0.2587 |
| GRU + Dense | MSE | 0.0666 | 0.0669 | **0.0665** | 0.0670 |
| | RMSE | 0.2580 | 0.2586 | 0.2578 | 0.2588 |
| Conv1D + Flatten + Dense | MSE | 0.1371 | 0.1747 | 0.1033 | **0.0980** |
| | RMSE | 0.3703 | 0.4180 | 0.3214 | 0.3130 |
| LSTM + Dropout + LSTM + Dense | MSE | **0.0684** | 0.0688 | 0.0686 | 0.0688 |
| | RMSE | 0.2615 | 0.2623 | 0.2619 | 0.2623 |
| SimpleRNN + LSTM + Dropout + LSTM + Dense | MSE | 0.0700 | 0.0706 | **0.0694** | 0.0712 |
| | RMSE | 0.2645 | 0.2657 | 0.2635 | 0.2669 |
| Conv1D + MaxPooling1D + Flatten + Dense | MSE | 0.1178 | 0.1362 | 0.1241 | **0.1087** |
| | RMSE | 0.3432 | 0.3691 | 0.3523 | 0.3297 |

PM10, followed by wind speed, solar elevation angle, and sea level pressure, among others (as depicted in Table 8).

## Features analysis

Each feature was analyzed individually and compared their impact on the other features. The Pearson correlation coefficient was computed to achieve this, and the values were compared,

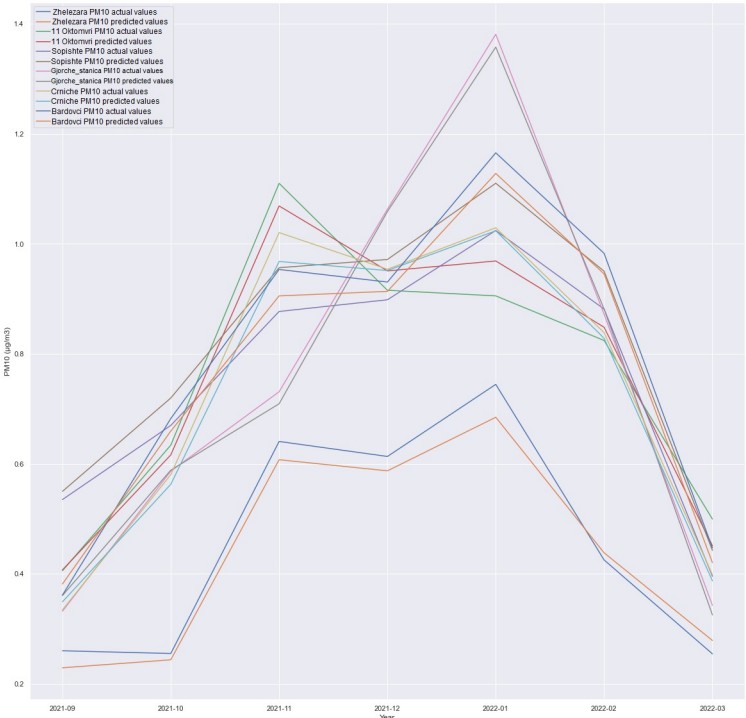

**Fig 6. GRU + Dense layer, forecast comparison for all locations with the 1h time horizon.**

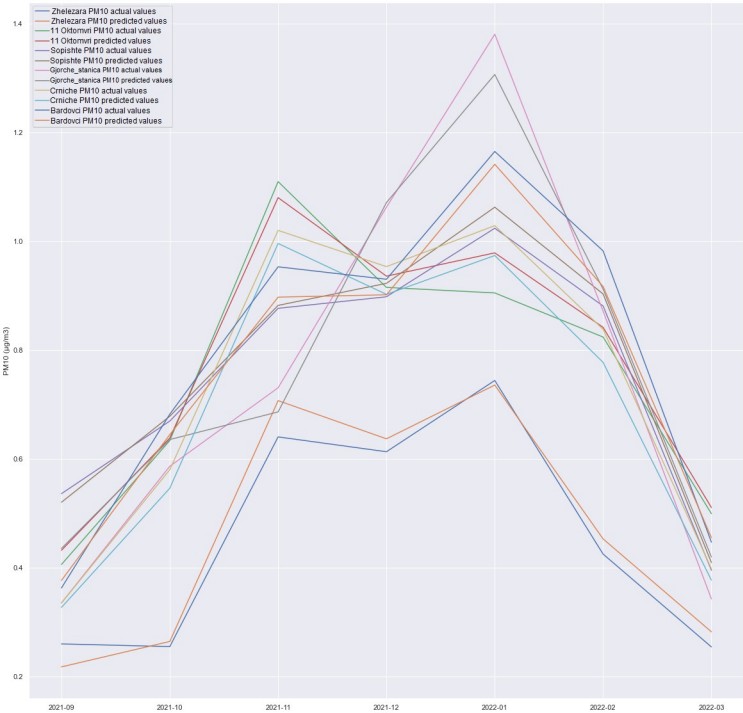

**Fig 7. GRU + Dense layer, forecast comparison for all locations with the 6h time horizon.**

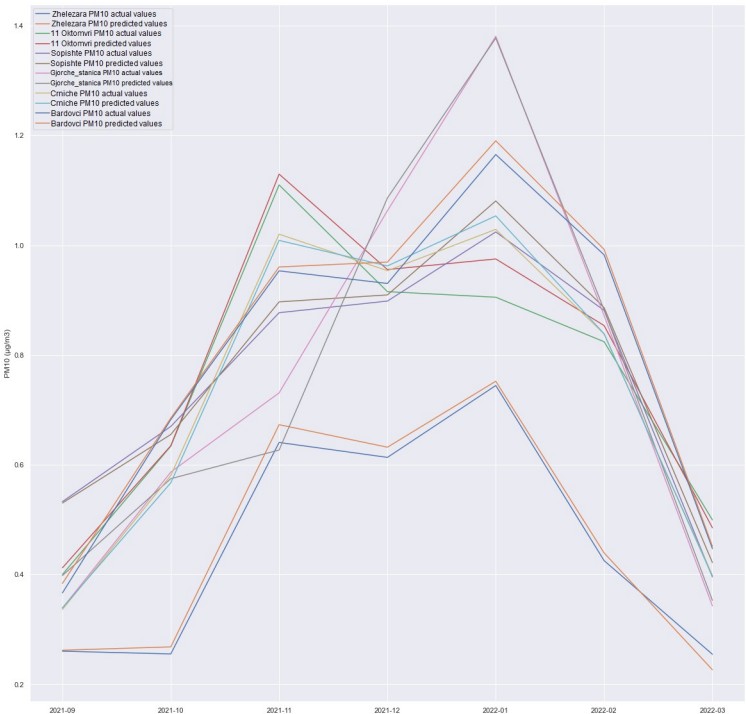

**Fig 8. GRU + Dense layer, forecast comparison for all locations with the 12h time horizon.**

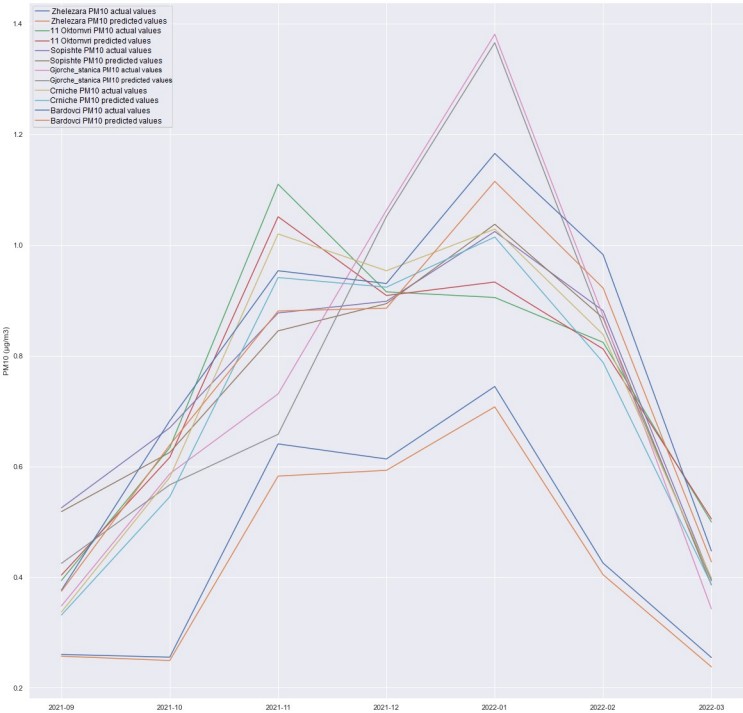

**Fig 9. GRU + Dense layer, forecast comparison for all locations with the 24h time horizon.**

as depicted in Table 9. Started by taking PM10 as a class and analyzing the other features related to PM10.

It was observed that temperature is slightly negatively correlated with PM10, which is unsurprising. Many studies have shown that pollution is higher during the winter months in colder conditions [44]. Therefore, higher temperatures lead to lower PM10 levels, as also evidenced by the visualizations. As the temperature increases, PM10 levels decrease, and vice versa. This was always true for all five locations.

The locations' noise's upper boundaries range from 160 to 200 decibels. It was noticed that noise is not strongly correlated with PM10, and the correlation coefficient is around 0.06 for all locations except for Gjorche Stanica, where the correlation is 0.23. However, this does not prove that noise impacts PM10 because, according to the data, the noise hit an all-time maximum in December 2018 for the dataset, and at the same time, the PM10 levels were high. This was expected for that period in Skopje, as it was a winter period, and the temperature was low. Therefore, it is assumed that this is a false-positive coefficient when drawing the final conclusion about whether noise impacts PM10 at all locations.

Humidity's correlation coefficient is the smallest for 11 Oktomvri and Bardovci, but they have more null values for this feature than the other locations. These values were later

**Table 7. A summary of the evaluated approaches along with the average forecasting performance, gauged through the MSE and RMSE, for multivariate outcomes.**

|                     | GRU+Dense(1h) | GRU+Dense(6h) | GRU+Dense(12h) | GRU+Dense(24h) |
| ------------------- | ------------- | ------------- | -------------- | -------------- |
| Activation Function | sigmoid       | sigmoid       | ReLU           | ReLU           |
| MSE                 | 0.1163        | 0.1321        | 0.1133         | 0.1130         |
| RMSE                | 0.3410        | 0.3635        | 0.3366         | 0.3361         |

**Table 8. Summary of the feature selection algorithms.**

| Feature | Algorithms | | | | | | Total |
|---|---|---|---|---|---|---|---|
| | *P | *C2 | *R | *L | *RF | *LGB | |
| PM25 | True | True | True | True | True | True | 6 |
| wind speed | True | True | True | False | False | True | 4 |
| solar elevation angle | True | True | True | False | False | True | 4 |
| sea level pressure | True | True | True | False | False | True | 4 |
| clouds | True | True | True | False | False | True | 4 |
| visibility | True | True | False | False | False | False | 3 |
| relative humidity | True | False | True | False | False | True | 3 |
| pressure | False | True | True | False | False | True | 3 |
| wind direction | False | False | True | False | False | True | 2 |
| UV | True | True | False | False | False | False | 2 |
| solar radiation | True | True | False | False | False | False | 2 |
| temperature | True | False | False | False | False | False | 1 |
| snow | False | True | False | False | False | False | 1 |
| noise | False | False | True | False | False | False | 1 |
| humidity | False | False | False | False | False | False | 0 |

*P = Pearson, *C2 = Chi-squared, *R = Recursive Feature Extraction, *L = LinearRegression, *RF = Random Forest, *LGB = LightGBM

**Table 9. Summary of the Pearson's coefficient values for PM10 and the other features.**

| Pearson's coefficient | | | | | |
|---|---|---|---|---|---|
| Feature | Coefficient | Feature | Coefficient | Feature | Coefficient |
| PM2.5 | 0.97 | UV | -0.2 | Pressure | 0.13 |
| Temperature | -0.41 | Sea level pressure | 0.2 | Noise | 0.06 |
| Visibility | -0.4 | Wind speed | -0.18 | Clouds | 0.05 |
| Solar elevation angle | -0.29 | Humidity | 0.15 | Wind direction | 0.03 |
| Solar radiation | -0.2 | Relative humidity | 0.15 | Snow | 0.02 |

interpolated, so it is necessary to exclude these locations to gain a broader perspective. Another thing that these locations have in common is that their upper boundaries hit 100, while the other three locations vary between 60 and 70. However, the other locations have a correlation coefficient of around 0.15, which is not significant enough to suggest that PM10 and humidity have a significant impact on each other.

PM2.5 is another pollutant particle that is often found alongside PM10. This feature has the highest correlation coefficient, which is around 0.97 for all locations. It has the same patterns as PM10, and has a pattern of multicollinearity. If the temperature is correlated with PM10, then the temperature is also correlated with PM2.5, and so on.

Relative humidity has the same impact but slightly different values than humidity. Therefore, the same conclusion that applies to humidity also applies to relative humidity.

Wind speed, measured in km/h, has a small impact on PM10 levels. Interestingly, the correlation coefficient is negative, suggesting that higher wind speeds lead to lower PM10 pollution. This dataset and the locations naturally do not witness very high wind speeds, which may be the reason for the small correlation coefficient. Further research with higher wind speeds is necessary.

Sea level pressure has a coefficient of correlation of about 0.2, and it fits the pattern for PM10. During the summer periods, the sea level pressure is lower, which is also true for PM10 (lower in summer), while in the winter, it's higher. Although the coefficient indicates that this feature is not strongly correlated with PM10, when compared to the other features, it can still be considered as an option when selecting the final features.

The solar elevation angle is linked with the sun's angle pointed out to the locations. As mentioned previously, during the summer periods, this angle is bigger than the angle in the winter periods, and this means that while the angle is bigger, the pollution is smaller, which explains the negative correlation coefficient.

Similar to the solar elevation angle, solar radiation is higher in summer periods compared to winter periods, implying that PM10 is lower in summer compared to winter. This can be easily noticed by just looking at the data.

Judging by the data, no pattern can be noticed that tells us whether the pressure is tied with PM10 or not. Pearson's coefficient is about 0.13, which is a positive number, indicating that there is only a slight correlation. Being positive means that the higher the pressure, the higher the PM10.

Snow, as a weather condition, is mostly linked with winter; however, snow as a feature has almost no impact on PM10. Since the distribution of snow is slightly bad, Pearson's coefficient is about 0.02. In the future, this feature could be excluded during the feature extraction process.

UV radiation naturally comes from the sun, meaning that a more powerful sun equals more UV radiation. The sun is much more powerful during the summer than in winter, meaning that the higher the UV, the lower the PM10 (and PM2.5 from multicollinearity).

In wind direction, depending on the geographical position of the locations, the coefficient of correlation varies between a positive correlation and a negative one. However, both of them are small values, meaning that wind direction has almost no impact on PM10.

Visibility is a good feature that has a slightly better coefficient of correlation compared to the other features. It has a negative correlation, meaning that the higher the visibility, the lower the PM10. This is expected and natural because higher pollution reduces visibility. Usually, higher pollution implies the formation of clouds of pollution that move through the city.

As seen from the data and the correlation coefficient, clouds have almost no impact on the PM10 particles.

## Quarantine COVID period

During this research, it was observed that the Pre-COVID period was more polluted than the COVID period, with an average ratio of approximately 1:3. However, when comparing the COVID quarantine in 2020 to the same period in 2019 when there was no COVID, the mean value of PM10 pollution did not differ significantly from the mean value of PM10 in 2020. This suggests that similar pollution patterns and values existed during the 2020 quarantine period as in 2019. The pollution levels in the post-COVID period were even lower than those during the COVID quarantine period, which is expected as the quarantine ended around May 20, 2020, marking the beginning of the summer period when pollution is typically lower than in winter periods.

Noise, however, does not differ much compared to the pre-COVID and COVID quarantine periods. It was noticed that the quarantine period had less noise, but usually in the range of 7 decibels. Interestingly enough, compared to last year's quarantine period of 2020, the noise is smaller and ranges again in 5–7 decibels.

**Table 10. Average values of PM10 and noise during the pre-COVID quarantine period, COVID quarantine period, post-COVID quarantine period, and the quarantine period.**

| Location | PM10 | | | | Noise | | | |
|---|---|---|---|---|---|---|---|---|
| | Pre | During | Post | Last Year | Pre | During | Post | Last Year |
| 11 Oktomvri | 77.76 | 19.70 | 8.71 | 19.11 | 59.53 | 52.32 | 62.76 | 61.54 |
| Bardovci | 40.27 | 17.98 | 12.45 | 17.32 | 28.81 | 28.19 | 28.41 | 36.81 |
| Crniche | 70.53 | 17.61 | 6.59 | 16.47 | 36.99 | 31.55 | 28.50 | 30.03 |
| Gjorche Stanica | 57.44 | 16.36 | 8.09 | 17.04 | 61.48 | 44.52 | 41.05 | 50.81 |
| Zhelezara | 37.61 | 10.44 | 5.69 | 10.11 | 43.06 | 38.88 | 37.89 | 43.93 |

The goal is to verify whether the noise and PM10 impact each other, i.e., whether higher noise means higher pollution. Table 10 depicts that the numbers represented by the mean show that even though the noise value is lower than 5–7 decibels, the pollution remained almost the same as the previous year, meaning it had no significant impact on it. The coefficient of correlation was computed to gain more knowledge about how linearly dependent noise and PM10 are for each location used in this experiment. The coefficient varies slightly, but it is generally close to zero. The values range from 0.05 to 0.089, which mathematically suggests that the dataset's noise and PM10 particles do not impact and influence each other.

## Discussion

A comparison of the performance between an LSTM-based architecture and an architecture consisting of a SimpleRNN and a Dense layer was conducted. Additionally, experiments involving the combination of SimpleRNN and LSTM layers were executed. Surprisingly, unlike the less complex architectures where LSTM and SimpleRNN were utilized independently, the combined architecture did not deliver superior results. The most promising outcomes were achieved using a straightforward GRU-based architecture followed by a Dense layer, as visualized in Figs 6 through 9. In this particular scenario, using CNN did not produce satisfactory results. Given the limited number of features, it was anticipated that adding additional layers could increase the risk of overfitting the data. Nevertheless, additional research and experimentation are necessary to validate this hypothesis.

Furthermore, although the suggested architectures generally exhibited good performance, they did not surpass the results obtained with GRU. This might be due to the considerable variability in meteorological conditions occurring over longer periods (e.g., significant changes in wind speed, rainfall, etc.), rendering predictions notably more challenging. It is imperative to conduct additional experiments incorporating extended input series to investigate whether the limitations of the 24-hour time horizon prediction are attributable to the shorter durations of input data.

The study comes with several limitations. Firstly, the absence of pollution and weather sensors within the same geographical location prevented the inclusion of geographical data in the analysis. Additionally, the study's scope was limited by the exclusion of certain weather parameters. Furthermore, the application of data interpolation within the training and validation sets might at times obscure pollution spikes, especially when the interpolation covers multiple hours of data. The results were generated through a division into training, validation, and test sets, which, while offering a comprehensive view of algorithm performance, it is susceptible to the random initialization of variables. An additional shortcoming of the study is that the data is limited to the area of the city of Skopje; additional data from other areas is needed to confirm

the efficiency and scalability of the proposed methods. Nevertheless, the study affirms that the stability of the results was verified by repeating the experiments on a full year of test data, which produced similar outcomes.

A potential avenue for future research could involve expanding the model to incorporate derived temporal features, incorporating variables such as the month of the year, weekday, holidays, and time of day. These supplementary features could provide valuable insights into factors such as traffic patterns and the operational hours of factories and industrial facilities. Furthermore, a denser sensor network would enhance the ability to pinpoint specific locations with elevated air pollution levels and potentially identify the sources of pollution.

## Conclusions

Consistent with the prior research, it is evident that meteorological parameters and measurements within the Skopje region exhibit a strong correlation with pollution levels. When integrated with historical PM10 and PM2.5 concentration data, these factors significantly enhance the predictive capabilities of the used models. Remarkably, this finding remains valid despite the considerable distance between pollution measurement stations and meteorological measurement stations. While this article did not delve into exploring the influence of geographical location within the Skopje city area on predictions, future research can utilize an expanded network of sensory data locations to monitor how air pollution evolves in various city areas. This approach may also shed light on potential sources of increased pollution.

The suggested architectures exhibit excellent performance and achieve precise short-term forecasts of PM10 concentrations. As expected, the model's performance diminishes with an extended time horizon due to the heightened complexity of predicting events that occur further into the future. The choice of RNNs and CNNs has again proven, as in previous examples in the literature, that given a multimodal time-series data, state-of-the-art results can be obtained by leveraging their predictive power and characteristics.

The architectures do not require feature selection, however the feature importance analysis in this study gives additional insights on how each of the features in the data potentially influence the target PM10 variable. As feature importance is a pre-modeling step, in future work, post-hoc explainers can be leveraged to see if these variables are also useful in the deep learning modeling approaches.

The incorporation of additional meteorological and air quality sensor stations holds the potential to improve result accuracy. Moreover, it could facilitate precise geographical location predictions, empowering authorities to implement targeted measures at specific locations rather than citywide interventions. The growing awareness of the pollution issue in Skopje is reflected in the installation of over 100 informal sensor stations, which are already providing valuable data. In future research, the goal is to leverage this informal information to integrate geographical location into the models, potentially improving their predictive capabilities. Additionally, plans include extending the prediction period from 24 hours to several days or weeks in advance, informing the public well in advance and giving authorities enough time to make informed decisions.

## Author Contributions

**Conceptualization:** Martin Kostadinov, Eftim Zdravevski, Petre Lameski, Biljana Stojkoska, Michael A. Herzog, Vladimir Trajkovik.

**Data curation:** Martin Kostadinov, Eftim Zdravevski.

**Formal analysis:** Martin Kostadinov, Biljana Stojkoska, Vladimir Trajkovik.

**Funding acquisition:** Biljana Stojkoska, Michael A. Herzog, Vladimir Trajkovik.

**Investigation:** Martin Kostadinov, Petre Lameski, Biljana Stojkoska.

**Methodology:** Eftim Zdravevski, Petre Lameski, Michael A. Herzog, Vladimir Trajkovik.

**Project administration:** Michael A. Herzog.

**Resources:** Vladimir Trajkovik.

**Software:** Martin Kostadinov, Eftim Zdravevski, Paulo Jorge Coelho.

**Supervision:** Petre Lameski, Michael A. Herzog, Vladimir Trajkovik.

**Validation:** Eftim Zdravevski, Petre Lameski, Paulo Jorge Coelho, Michael A. Herzog, Vladimir Trajkovik.

**Visualization:** Martin Kostadinov, Eftim Zdravevski.

**Writing – original draft:** Martin Kostadinov, Eftim Zdravevski, Petre Lameski, Paulo Jorge Coelho, Michael A. Herzog, Vladimir Trajkovik.

**Writing – review & editing:** Martin Kostadinov, Eftim Zdravevski, Petre Lameski, Paulo Jorge Coelho, Biljana Stojkoska, Michael A. Herzog, Vladimir Trajkovik.

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
