## [Decision Letter · Decision Letter 0]

28 May 2024

PONE-D-24-05065Forecasting Air Pollution with Deep Learning with a focus on Impact of Urban Traffic on Air and Noise PollutionPLOS ONE

Dear Dr. Lameski,

Thank you for submitting your manuscript to PLOS ONE. After careful consideration, we feel that it has merit but does not fully meet PLOS ONE’s publication criteria as it currently stands. Therefore, we invite you to submit a revised version of the manuscript that addresses the points raised during the review process.

The three reviewers have agreed that your papers needed substantial improvements in many aspects. Please address these comments and read your manuscript carefully. There are many statements or claims remain unclear, specially the ones related to the experimental settings and your proposed methodology.

We look forward to receiving your revised manuscript.

Kind regards,

Amgad Muneer

Academic Editor

PLOS ONE

Journal Requirements:

[The work presented in this article was partially funded by the Ss. Cyril and Methodius

University in Skopje, Faculty of Computer Science and Engineering. We also

acknowledge the support of NVIDIA through a donation of a Titan V GPU. E.Z., P.L.,

M.K., B.S., V.T. and M.H acknowledge the support of the CleanBreathe project. P.J.C.

acknowledges the funding by FCT/MEC through national funds and, when applicable,

co-funded by the FEDER-PT2020 partnership agreement under the project

UIDB/00308/2020.]

 [The projects funding the research are included in the acknowledgement section of the publication.]

5. We note that you have indicated that there are restrictions to data sharing for this study. PLOS only allows data to be available upon request if there are legal or ethical restrictions on sharing data publicly. For more information on unacceptable data access restrictions, please see http://journals.plos.org/plosone/s/data-availability#loc-unacceptable-data-access-restrictions.

Reviewers' comments:

Reviewer's Responses to Questions

**Comments to the Author**

1. Is the manuscript technically sound, and do the data support the conclusions?

Reviewer #1: Partly

Reviewer #2: Partly

Reviewer #3: Partly

2. Has the statistical analysis been performed appropriately and rigorously? 

Reviewer #1: Yes

Reviewer #2: No

Reviewer #3: N/A

3. Have the authors made all data underlying the findings in their manuscript fully available?

Reviewer #1: No

Reviewer #2: No

Reviewer #3: Yes

4. Is the manuscript presented in an intelligible fashion and written in standard English?

Reviewer #1: Yes

Reviewer #2: No

Reviewer #3: Yes

5. Review Comments to the Author

Reviewer #1: Dear Authors,

I have carefully reviewed your manuscript and would like to offer some constructive feedback to enhance its quality and clarity. This study introduces a novel approach to forecasting air pollution in Skopje by leveraging Recurrent Neural Networks (RNNs) with Long Short-Term Memory (LSTM) units, focusing on PM10 particle levels across five city locations. By analyzing historical air quality data alongside meteorological conditions, and comparing various deep learning model implementations, the research aims to enhance prediction accuracy. Additionally, it investigates the impact of urban traffic on air and noise pollution, uniquely utilizing the COVID-19 lockdown period as a case study to assess traffic's role. The findings suggest that urban traffic is not the primary contributor to air pollution, marking a significant step in understanding environmental influences in urban settings.

Below are my suggestions by line number:

65: The organization of the manuscript is not well-written at the end of the introduction.

86: Citation 20 should be replaced by the original paper of LSTM:

Hochreiter, S., & Schmidhuber, J. (1997). Long short-term memory. Neural computation, 9(8), 1735-1780.

140: You wrote, "this research takes into account PM2.5 values", while in your abstract PM10 was written. Hence, which one was taken into account.

159: "The validation samples were dynamically selected, constituting 20 percent of the training data points.", it's advice to write how many samples?

160: "Hyperparameter tuning was done manually on the data for November 11, 2019, following the approaches and methods in [6]"

I checked the cited paper, and it stats that Keras Tuner library was utilized to optimze their proposed model, along with the manual selection of data for validation. The hyperparmeters tuned in that paper were learning rate, dropout rate, layers units, kernal size, num. of filters. etc. You can mention Keras Tuner as well with more details.

210: add the equation for MSE, and you should consider other evaluation metrics as well.

256: add supporting citation for the first paragraph that defines the feature selection process.

260: "we explore three popular feature selection methods: Pearson’s correlation coefficient, Chi-squared test, and K best features from Linear Regression, RandomForest Regression, and LightGBM",

263: add supporting citation for each technique (Pearson’s correlation coefficient, Chi-squared test, and K best features).

320: RMSE is used while it's not written in the line 210, if you plan to use it, then add its equation.

363: "Many studies have shown that pollution is higher during the winter months in colder conditions". Add referecnes to those studies.

The conclusion needs to add more details about the conducted experiment as it is now only summerize the limitations and future works that needs to be done.

In general, the paper needs to add more references to support its claims.

Figure 1: It is a trivial and very basic figure which doesn't reflect the methedology of the manuscript. You should redesign it using programs like draw.io or any alternative.

Increate the resolutions of your figures as I couldn't read their data.

It is advisable to avoid the first-person perspective in academic writing. Consider using passive voice constructions instead of "we" and "our."

You should add the architectures of the used techniques . i.e. RNN, LSTM, GRU, CNN etc. so novice readers can understand your paper.

I hope these comments are helpful in refining your manuscript. Your efforts to advance the understanding of fake news detection using deep learning techniques are commendable, and I look forward to the revised version of your work.

Best regards

Reviewer #2: The central idea of the article titled “Forecasting Air Pollution with Deep Learning with a focus on Impact of Urban Traffic on Air and Noise Pollution” is well-conceived and contributes valuable insights to the field of pollution. However, the paper would benefit from improved organization and clarity. Here are my suggestions:

1. The title of the article, “Forecasting Air Pollution with Deep Learning with a focus on Impact of Urban Traffic on Air and Noise Pollution,” is somewhat misleading. The term "pollution" is very general and encompasses a wide range of pollutants. To be more precise and reflective of the content, the title should specify that the study focuses on forecasting specific pollutants, such as PM2.5 and PM10. This will provide clarity to the readers about the scope of your research.

2. The introduction of the article lacks a clear structure, making it difficult to follow the main themes. At times, the author discusses works related to pollution, while at other times, the focus shifts to machine learning and IoT. To improve the flow and coherence, I suggest organizing the introduction into distinct sections. Start with an overview of the pollution problem, followed by a review of relevant works on pollution. Then, discuss the role of machine learning and IoT in addressing these issues, clearly linking each section to the overarching aim of the study.

3. The statement, "The primary innovation of our study lies in the utilization of diverse datasets obtained from local sensors to enhance prediction accuracy," suggests a novel approach. However, it is important to note that many existing works have already utilized similar methods. Therefore, this aspect of the study may not be as innovative as claimed. I recommend highlighting what specifically differentiates your approach from these existing works to clearly demonstrate its unique contributions.

4. The phrase, "An idea was put forth to forecast PM10 concentration for various time intervals by employing three distinct stepwise Multiple Linear Regression (MLR) models as proposed in [17]. In [18], the authors forecasted the concentration of a heavily polluted area," is not well expressed. It would benefit from clearer structure and coherence. For instance, consider rephrasing it to: "The study proposed forecasting PM10 concentrations for various time intervals using three distinct stepwise Multiple Linear Regression (MLR) models, as outlined in [17]. Additionally, [18] focused on forecasting the concentration of pollutants in a heavily polluted area."

5. While the authors cited the measurements gathered at each location, it is noted elsewhere in the article that meteorological variables are also included. To present the data with greater clarity, we recommend that the authors include a table that lists all the features, including the meteorological variables, along with their respective time intervals.

6. In the feature selection paragraph, the authors use the chi-squared (chi2) method for feature selection. However, since the authors are working with numerical values and the chi-squared method is designed only for categorical variables, this approach is not appropriate for the data being used.

7. The authors state that "Feature selection is a crucial step in building effective and efficient machine learning models. The process entails identifying and preserving the most pertinent features from the dataset to enhance model performance while also reducing computational complexity." However, it appears that the feature selection process described was not effectively utilized to enhance the model's performance or reduce computational complexity. More details on how the selected features improved the model and reduced complexity would be beneficial.

8. The purpose of this work is to predict the concentration of PM2.5 and PM10 for 1, 6, 12, and 24 hours ahead. However, I have concerns about the practical benefits of these short-term predictions given the methodology used. The model relies on measurements taken every 15 minutes over the last day. Therefore, if we already have detailed concentration data for each 15-minute interval from the previous day, it is unclear how beneficial it is to predict concentrations for the next 1, 6, 12, or 24 hours, as these predictions may closely mirror the short-term past data. It might be more useful to focus on predicting the concentration of these pollutants over a longer period, such as a week or a month, to provide more valuable insights for planning and intervention.

Reviewer #3: This study presents an LSTM and RNN approach to address the challenge of food supply purity and well-being. The topic is relevant and of significant interest to researchers. The contribution of the authors mainly lies in the data collection part. However, the study can benefit from the following comments:

Points for Improvement

The problem statement should be derived from the literature. I suggest citing relevant studies in the introduction to strengthen the claims of the authors.

The authors should clearly mention why LSTM and CNN are selected. What are the rationales behind selecting these models? Discussing the potential impact of the study in both the introduction and conclusion will showcase the importance of the study.

Data Description

The description, size, and dimensions of the data are not clearly described. Th

The results should be discussed rather than just presented. Additionally, it is not clear why the chosen evaluation measures are selected. The authors should explain the choice of these metrics.

The authors are advised to perform a gap analysis among recent studies. This will help in identifying the unique contributions of the current study and how it advances the field.

Real-World Applicability

Are the methods and findings applicable to real-world air pollution detection scenarios? The authors should discuss the practical implications and applicability of their approach.

Scalability

Is the approach scalable to different datasets or geographical areas? This is important to understand the broader applicability of the study’s findings.

Authors can discuss the limitations of the study and how addressing them is important.

A recent helpful study related to CNN and LSTM can be beneficial for the authors

Abid, Yawar Abbas, Jinsong Wu, Guangquan Xu, Shihui Fu, and Muhammad Waqas. "Multi-Level Deep Neural Network for Distributed Denial-of-Service Attack Detection and Classification in Software-Defined Networking Supported Internet of Things Networks." IEEE Internet of Things Journal (2024).

This will add credibility to the research.

6. PLOS authors have the option to publish the peer review history of their article (what does this mean?). If published, this will include your full peer review and any attached files.

Reviewer #1: No

Reviewer #2: **Yes: **Samira Douzi

Reviewer #3: No

---

## [Author Response · Author response to Decision Letter 0]

12 Jul 2024

Response to reviewers:

Response to Reviewer#1:

We thank the reviewer for their constructive feedback and suggestions. Please see the responses below:

Reviewer #1: Dear Authors,

I have carefully reviewed your manuscript and would like to offer some constructive feedback to enhance its quality and clarity. This study introduces a novel approach to forecasting air pollution in Skopje by leveraging Recurrent Neural Networks (RNNs) with Long Short-Term Memory (LSTM) units, focusing on PM10 particle levels across five city locations. By analyzing historical air quality data alongside meteorological conditions, and comparing various deep learning model implementations, the research aims to enhance prediction accuracy. Additionally, it investigates the impact of urban traffic on air and noise pollution, uniquely utilizing the COVID-19 lockdown period as a case study to assess traffic's role. The findings suggest that urban traffic is not the primary contributor to air pollution, marking a significant step in understanding environmental influences in urban settings.

Below are my suggestions by line number:

65: The organization of the manuscript is not well-written at the end of the introduction.

Response: We have addressed the issue at the end of the introduction section. The paper structure now is written, and the referencing errors fixed.

86: Citation 20 should be replaced by the original paper of LSTM:

Hochreiter, S., & Schmidhuber, J. (1997). Long short-term memory. Neural computation, 9(8), 1735-1780.

Response: We have replaced the reference with the proper one.

140: You wrote, "this research takes into account PM2.5 values", while in your abstract PM10 was written. Hence, which one was taken into account.

Response: We take both into account but we target only PM10. The sentence is rephrased to be more clear:

In contrast to those methodologies, this research additionally considers PM2.5 values and the concentrations of additional meteorological variables at the measurement stations in Skopje, which have not been examined previously. The locations in Skopje where the data was collected data are listed in Table 1

159: "The validation samples were dynamically selected, constituting 20 percent of the training data points.", it's advice to write how many samples?

Response: We added the information about the number of samples. Here is the changed text:

The dataset was partitioned into separate training, validation, and test sets. The training data encompassed the period from September 11, 2017, to February 3, 2022, amounting to a total of 151,199 samples. The validation samples were dynamically selected, constituting 20 percent of the training data points amounting to a total of 30,239 samples.

160: "Hyperparameter tuning was done manually on the data for November 11, 2019, following the approaches and methods in [6]"

I checked the cited paper, and it stats that Keras Tuner library was utilized to optimze their proposed model, along with the manual selection of data for validation. The hyperparmeters tuned in that paper were learning rate, dropout rate, layers units, kernal size, num. of filters. etc. You can mention Keras Tuner as well with more details.

Response: We have mentioned the Keras Tuner in the text and added appropriate reference.

210: add the equation for MSE, and you should consider other evaluation metrics as well.

Response: We have added the MSE and RMSE metrics equation in the subsection Methods Evaluation.

256: add supporting citation for the first paragraph that defines the feature selection process.

Response: We have added a supporting citation.

260: "we explore three popular feature selection methods: Pearson’s correlation coefficient, Chi-squared test, and K best features from Linear Regression, RandomForest Regression, and LightGBM",

263: add supporting citation for each technique (Pearson’s correlation coefficient, Chi-squared test, and K best features).

Response: We have added supporting citations for each technique.

320: RMSE is used while it's not written in the line 210, if you plan to use it, then add its equation.

Response: We have added the equation for RMSE.

363: "Many studies have shown that pollution is higher during the winter months in colder conditions". Add referecnes to those studies.

Response: We have added an appropriate reference.

The conclusion needs to add more details about the conducted experiment as it is now only summerize the limitations and future works that needs to be done.

In general, the paper needs to add more references to support its claims.

Figure 1: It is a trivial and very basic figure which doesn't reflect the methedology of the manuscript. You should redesign it using programs like draw.io or any alternative.

Increate the resolutions of your figures as I couldn't read their data.

Response: Thank you for the suggestion. We have improved the figure.

It is advisable to avoid the first-person perspective in academic writing. Consider using passive voice constructions instead of "we" and "our."

You should add the architectures of the used techniques . i.e. RNN, LSTM, GRU, CNN etc. so novice readers can understand your paper.

I hope these comments are helpful in refining your manuscript. Your efforts to advance the understanding of fake news detection using deep learning techniques are commendable, and I look forward to the revised version of your work.

Best regards

Response: We thank the reviewer for the detailed suggestions. We have addressed them and the changes are reflected in the new version of the article.

Response to Reviewer#2:

We thank the reviewer for the constructive feedback and the suggestions to improve the article. Please find the responses below:

Reviewer #2: The central idea of the article titled “Forecasting Air Pollution with Deep Learning with a focus on Impact of Urban Traffic on Air and Noise Pollution” is well-conceived and contributes valuable insights to the field of pollution. However, the paper would benefit from improved organization and clarity. Here are my suggestions:

1. The title of the article, “Forecasting Air Pollution with Deep Learning with a focus on Impact of Urban Traffic on Air and Noise Pollution,” is somewhat misleading. The term "pollution" is very general and encompasses a wide range of pollutants. To be more precise and reflective of the content, the title should specify that the study focuses on forecasting specific pollutants, such as PM2.5 and PM10. This will provide clarity to the readers about the scope of your research.

Response: We thank the reviewer for the suggestion. We have amended the title of the publication: Forecasting Air Pollution with Deep Learning with a focus on Impact of Urban Traffic on PM10 and Noise Pollution

2. The introduction of the article lacks a clear structure, making it difficult to follow the main themes. At times, the author discusses works related to pollution, while at other times, the focus shifts to machine learning and IoT. To improve the flow and coherence, I suggest organizing the introduction into distinct sections. Start with an overview of the pollution problem, followed by a review of relevant works on pollution. Then, discuss the role of machine learning and IoT in addressing these issues, clearly linking each section to the overarching aim of the study.

Response: We thank the reviewer for this suggestion. We have reorganized the Introduction section by grouping the pollution and health related references at the beginning and then continuing with the IoT, Statistics and ML.

3. The statement, "The primary innovation of our study lies in the utilization of diverse datasets obtained from local sensors to enhance prediction accuracy," suggests a novel approach. However, it is important to note that many existing works have already utilized similar methods. Therefore, this aspect of the study may not be as innovative as claimed. I recommend highlighting what specifically differentiates your approach from these existing works to clearly demonstrate its unique contributions.

Response: We thank the reviewer for this remark. We improved the text based on this:

In this paper, we utilize air quality measurements along with meteorological data to predict levels of air pollution in the urban region of Skopje for time intervals of 1, 6, 12, and 24 hours. The primary innovation of our study lies in the utilization of diverse datasets obtained from local sensors to enhance prediction accuracy. Furthermore, we compile historical data from meteorological stations in the vicinity of Skopje. We assess the performance of various models employing Long Short-Term Memory (LSTM) networks and Convolutional Neural Networks (CNNs) and conduct a comparative analysis of their effectiveness. Additional contribution is the newely compiled and preprocessed dataset that is made publicly available with this study and the applied methods implementation and the presented results and the identification of the influence of traffic to the PM10 pollution in the area of Skopje.

4. The phrase, "An idea was put forth to forecast PM10 concentration for various time intervals by employing three distinct stepwise Multiple Linear Regression (MLR) models as proposed in [17]. In [18], the authors forecasted the concentration of a heavily polluted area," is not well expressed. It would benefit from clearer structure and coherence. For instance, consider rephrasing it to: "The study proposed forecasting PM10 concentrations for various time intervals using three distinct stepwise Multiple Linear Regression (MLR) models, as outlined in [17]. Additionally, [18] focused on forecasting the concentration of pollutants in a heavily polluted area."

Response: We thank the reviewer for the suggestion. We have used it to improve the text.

5. While the authors cited the measurements gathered at each location, it is noted elsewhere in the article that meteorological variables are also included. To present the data with greater clarity, we recommend that the authors include a table that lists all the features, including the meteorological variables, along with their respective time intervals.

Response: The meteorological variables used for building the features and models are given in Table 8: Summary of the feature selection algorithms

6. In the feature selection paragraph, the authors use the chi-squared (chi2) method for feature selection. However, since the authors are working with numerical values and the chi-squared method is designed only for categorical variables, this approach is not appropriate for the data being used.

Response: We thank the reviewer for reminding us of the limitation of the chi-squared method. All of the variables used in the analysis are numerical.

7. The authors state that "Feature selection is a crucial step in building effective and efficient machine learning models. The process entails identifying and preserving the most pertinent features from the dataset to enhance model performance while also reducing computational complexity." However, it appears that the feature selection process described was not effectively utilized to enhance the model's performance or reduce computational complexity. More details on how the selected features improved the model and reduced complexity would be beneficial.

Response: We thank the reviewer for the very helpful remark. We have improved the text to be more clear about why we include the feature selection. In the Feature analysis and selection section and in the Conclusion.

8. The purpose of this work is to predict the concentration of PM2.5 and PM10 for 1, 6, 12, and 24 hours ahead. However, I have concerns about the practical benefits of these short-term predictions given the methodology used. The model relies on measurements taken every 15 minutes over the last day. Therefore, if we already have detailed concentration data for each 15-minute interval from the previous day, it is unclear how beneficial it is to predict concentrations for the next 1, 6, 12, or 24 hours, as these predictions may closely mirror the short-term past data. It might be more useful to focus on predicting the concentration of these pollutants over a longer period, such as a week or a month, to provide more valuable insights for planning and intervention.

Response: We thank the reviewer for this suggestion. We have decided to perform the longer period predictions for future work as stated in the Conclusion.

Response to Reviewer#3:

We thank the reviewer for the constructive feedback, the suggestions and the remarks. Please find the responses below:

Reviewer #3: This study presents an LSTM and RNN approach to address the challenge of food supply purity and well-being. The topic is relevant and of significant interest to researchers. The contribution of the authors mainly lies in the data collection part. However, the study can benefit from the following comments:

Points for Improvement

The problem statement should be derived from the literature. I suggest citing relevant studies in the introduction to strengthen the claims of the authors.

Response: We thank the reviewers suggestion. We have improved the introduction section accordingly.

The authors should clearly mention why LSTM and CNN are selected. What are the rationales behind selecting these models? Discussing the potential impact of the study in both the introduction and conclusion will showcase the importance of the study.

Response: We thank the reviewer for the question and the recommendation. Our dataset is a time series dataset so that’s the main reason for choosing LSTM. LSTMs provide a powerful framework for modeling time series data due to their ability to learn and remember long-term dependencies, capture complex temporal patterns, and handle the sequential nature of the data. Their flexibility and robustness make them a go-to choice for a wide range of time series applications, from forecasting to anomaly detection. The reason for choosing CNN is because we perform a sliding-window approach where we basically “group” a chunk of data for training and have the next sample for prediction. CNNs excel at detecting local patterns and features within data. For time series, this means they can identify important temporal patterns, trends, and anomalies within each window of data. The convolutional layers automatically learn the most relevant features from the time series data, reducing the need for manual feature engineering. We have improved the text by describing this in more details in subsection: The architecture of Deep Learning Models:

The reason for choosing these types of deep learning models is that the dataset contains time series data. RNNs have proven to be good at learning time series data characteristics and patterns, while CNNs are able to detect local features and patterns in the data even for time series. Both approaches are able to generate representative features from time series data and have proven to build models which are good at forecasting values.

In the conclusion:

The suggested architectures exhibit excellent performance and achieve precise short-term forecasts of PM10 concentrations. As expected, the model's performance diminishes with an extended time horizon due to the heightened complexity of predicting events that occur further into the future. The choice of RNNs and CNNs have again proven as in previous examples in the literature that given a multimodal time-series data, like in our case, we can obtain state-of-the-art results by leveraging their predictive power and characteristics.

Data Description

The description, size, and dimensions of the data are not clearly described. Th

Response: We have included more information about the dataset including the number of samples in the text.

The results should be discussed rather than just presented. Additionally, it is not clear why the chosen evaluation measures are selected. The authors should explain the choice of these metrics.

Response: The meteorological variables used for building the features and models are given in Table 8: Summary of the feature selection algorithms

The authors are advised to perform a gap analysis among recent studies. This will help

---

## [Decision Letter · Decision Letter 1]

5 Aug 2024

PONE-D-24-05065R1Forecasting Air Pollution with Deep Learning with a focus on Impact of Urban Traffic on PM10 and Noise PollutionPLOS ONE

Dear Dr. Lameski,

Thank you for submitting your manuscript to PLOS ONE. After careful consideration, we feel that it has merit but does not fully meet PLOS ONE’s publication criteria as it currently stands. Therefore, we invite you to submit a revised version of the manuscript that addresses the points raised during the review process.

We look forward to receiving your revised manuscript.

Kind regards,

Amgad Muneer

Academic Editor

PLOS ONE

Journal Requirements:

Reviewers' comments:

Reviewer's Responses to Questions

**Comments to the Author**

1. If the authors have adequately addressed your comments raised in a previous round of review and you feel that this manuscript is now acceptable for publication, you may indicate that here to bypass the “Comments to the Author” section, enter your conflict of interest statement in the “Confidential to Editor” section, and submit your "Accept" recommendation.

Reviewer #1: (No Response)

Reviewer #3: All comments have been addressed

2. Is the manuscript technically sound, and do the data support the conclusions?

Reviewer #1: Yes

Reviewer #3: Partly

3. Has the statistical analysis been performed appropriately and rigorously? 

Reviewer #1: Yes

Reviewer #3: N/A

4. Have the authors made all data underlying the findings in their manuscript fully available?

Reviewer #1: (No Response)

Reviewer #3: No

5. Is the manuscript presented in an intelligible fashion and written in standard English?

Reviewer #1: Yes

Reviewer #3: Yes

6. Review Comments to the Author

Reviewer #1: Thank you for your revised submission. I have reviewed your manuscript again and you have addressed all my previous comments, but I have four minor concerns for improvement. Please address these points and submit the revised version.

Concern 1:

It is advisable to avoid the first-person perspective in academic writing. Consider using passive voice constructions instead of "we" and "our."

Concern 2:

No need to repeat the full abbreviations multiple times. Since you've already declared them once, use the abbreviations consistently throughout the text. I've noticed many of them.

Concern 3:

342:

"we chose these metrics because MSE and RMSE are robust and informative metrics for evaluating regression model".

Instead of the original sentence, you may write: "MSE and RMSE are among the most commonly used evaluation metrics for regression models, as noted in recent systematic literature reviews"

Al-Selwi, S. M., Hassan, M. F., et al. (2024). RNN-LSTM: From applications to modeling techniques and beyond—Systematic review. Journal of King Saud University-Computer and Information Sciences, 102068.

Concern 4:

282:

Avoid using contractions like "don't" in academic writing. Instead, use the full form, such as "do not," to maintain a formal tone.

I look forward to your updated submission.

Reviewer #3: (No Response)

7. PLOS authors have the option to publish the peer review history of their article (what does this mean?). If published, this will include your full peer review and any attached files.

Reviewer #1: No

Reviewer #3: No

---

## [Author Response · Author response to Decision Letter 1]

9 Sep 2024

Response to Reviewer#1:

We thank the reviewer for their constructive feedback and suggestions. Please see the responses below:

Reviewer #1: Thank you for your revised submission. I have reviewed your manuscript again and you have addressed all my previous comments, but I have four minor concerns for improvement. Please address these points and submit the revised version.

Concern 1:

It is advisable to avoid the first-person perspective in academic writing. Consider using passive voice constructions instead of "we" and "our."

Response: We have rephrased the sentences as suggested.

Concern 2:

No need to repeat the full abbreviations multiple times. Since you've already declared them once, use the abbreviations consistently throughout the text. I've noticed many of them.

Response: We have fixed this issue throughout the manuscript.

Concern 3:

342:

"we chose these metrics because MSE and RMSE are robust and informative metrics for evaluating regression model".

Instead of the original sentence, you may write: "MSE and RMSE are among the most commonly used evaluation metrics for regression models, as noted in recent systematic literature reviews"

Al-Selwi, S. M., Hassan, M. F., et al. (2024). RNN-LSTM: From applications to modeling techniques and beyond—Systematic review. Journal of King Saud University-Computer and Information Sciences, 102068.

Response: We have modified the manuscript according to this and included the suggested reference.

Concern 4:

282:

Avoid using contractions like "don't" in academic writing. Instead, use the full form, such as "do not," to maintain a formal tone.

Response: We have modified the article according to this suggestion.

---

## [Editor Report · Decision Letter 2]

23 Oct 2024

Forecasting Air Pollution with Deep Learning with a focus on Impact of Urban Traffic on PM10 and Noise Pollution

PONE-D-24-05065R2

Dear Dr. Petre Lameski,

We’re pleased to inform you that your manuscript has been judged scientifically suitable for publication and will be formally accepted for publication once it meets all outstanding technical requirements (the figures provided are blurry, please ensure to provide them with better quality before publication).

Kind regards,

Amgad Muneer

Academic Editor

PLOS ONE

Additional Editor Comments (optional):

The authors have addressed all the reviewers comments and paper ready for acceptance except all the figures are blurry and authors needs to provide a better quality figure before publication
---

## [Editor Report · Acceptance letter]

28 Nov 2024

PONE-D-24-05065R2 

PLOS ONE

Dear Dr. Lameski, 

I'm pleased to inform you that your manuscript has been deemed suitable for publication in PLOS ONE. Congratulations! Your manuscript is now being handed over to our production team.

Kind regards, 

on behalf of

Dr. Amgad Muneer 

Academic Editor

PLOS ONE